# Double-Fermented Soybean Meal Totally Replaces Soybean Meal in Broiler Rations with Favorable Impact on Performance, Digestibility, Amino Acids Transporters and Meat Nutritional Value

**DOI:** 10.3390/ani13061030

**Published:** 2023-03-11

**Authors:** Sherief M. Abdel-Raheem, El Said Yehia Mohammed, Rania Elsaid Mahmoud, Mahmoud Fathy El Gamal, Hend S. Nada, Waleed Rizk El-Ghareeb, Mohamed Marzok, Ahmed M. A. Meligy, Mohamad Abdulmohsen, Hesham Ismail, Doaa Ibrahim, Asmaa T. Y. Kishawy

**Affiliations:** 1Department of Public Health, College of Veterinary Medicine, King Faisal University, Al-Ahsa, Hofuf 31982, Saudi Arabia; 2Department of Animal Nutrition and Clinical Nutrition, Faculty of Veterinary Medicine, Assiut University, Assiut 71526, Egypt; 3Department of Nutrition and Clinical Nutrition, Faculty of Veterinary Medicine, Zagazig University, Zagazig 44519, Egypt; 4Department of Microbiology, Faculty of Veterinary Medicine, Zagazig University, Zagazig 44519, Egypt; 5Food Control Department, Faculty of Veterinary Medicine, Zagazig University, Zagazig 44519, Egypt; 6Department of Clinical Sciences, College of Veterinary Medicine, King Faisal University, Al-Ahsa, Hofuf 31982, Saudi Arabia; 7Department of Surgery, Faculty of Veterinary Medicine, KafrelSheikh University, KafrelSheikh 33511, Egypt; 8Department of Physiology, Agricultural Research Center (ARC), Giza 13611, Egypt; 9Department of Animal Behavior, Faculty of Veterinary Medicine, Suez Canal University, Ismailia 41522, Egypt; 10Food Hygiene Department, Faculty of Veterinary Medicine, Assiut University, Assiut 71526, Egypt

**Keywords:** double-fermented soybean meal, digestive enzymes, amino acid transporters, phytase, broiler chicken

## Abstract

**Simple Summary:**

The presence of anti-nutritional factors (ANFs) in soybean meal (SBM) is considered the main question that motivates the poultry feed industry to develop its traditional processing techniques. Phytate and trypsin inhibitors are the prominent ANFs in SBM that inhibit nutrient digestion and absorption. One of the most recent and effective processing methods is fermentation of SBM with different fermentative microorganisms. Applying double stages of microbial fermentation of SBM, utilizing *Asperigillus oryzae* and *Bacillus subtilis*, was proven to be a highly effective method in eradication of most ANFs, such as phytate and trypsin inhibitors, while simultaneously improving SBM protein and amino acid content. Inclusion of double-fermented soybean meal (DFSBM) in broiler rations triggered better feed efficiency, nutrient digestibility, and amino acid transporters that improved the birds’ weight gain and muscle nutritional value.

**Abstract:**

Inclusion of microbial fermented soybean meal in broiler feed has induced advantageous outcomes for their performance and gastrointestinal health via exhibiting probiotic effects. In this study, soybean meal (SBM) was subjected to double-stage microbial fermentation utilizing functional metabolites of fungi and bacteria. In broiler diet, DFSBM replaced SBM by 0, 25, 50 and 100%. DFSBM was reported to have higher protein content and total essential, nonessential and free amino acids (increased by 3.67%, 12.81%, 10.10% and 5.88-fold, respectively, compared to SBM). Notably, phytase activity and lactic acid bacteria increased, while fiber, lipid and trypsin inhibitor contents were decreased by 14.05%, 38.24% and 72.80%, respectively, in a diet containing 100% DFSBM, compared to SBM. Improved growth performance and apparent nutrient digestibility, including phosphorus and calcium, and pancreatic digestive enzyme activities were observed in groups fed higher DFSBM levels. In addition, higher inclusion levels of DFSBM increased blood immune response (IgG, IgM, nitric oxide and lysozyme levels) and liver antioxidant status. Jejunal amino acids- and peptide transporter-encoding genes (LAT1, CAT-1, CAT-2, PepT-1 and PepT-2) were upregulated with increasing levels of DFSBM in the ration. Breast muscle crude protein, calcium and phosphorus retention were increased, especially at higher inclusion levels of DFSBM. Coliform bacteria load was significantly reduced, while lactic acid bacteria count in broiler intestines was increased with higher dietary levels of DFSBM. In conclusion, replacement of SBM with DFSBM positively impacted broiler chicken feed utilization and boosted chickens’ amino acid transportation, in addition to improving the nutritional value of their breast meat.

## 1. Introduction

Soybean meal (SBM), a co-product of oil refining from soy seed, is the most nutritious source of plant protein in the poultry industry. However, use of this byproduct requires further processing to avoid its anti-nutritional factors (ANFs), such as trypsin inhibitors, oligosaccharides and allergenic proteins, which reduce digestion, absorption and utilization of other essential nutrients, which has harmful impacts on animal health [1]. SBM phytate can bind carbohydrates, amino acids, microminerals and a variety of digestive enzymes, which in turn hinders the absorption of nutrients in broiler diets, as endogenous phytase is not secreted in the chicken’s gut [2]. Additionally, more than half of the total phosphorus in SBM is present as part of phytate that is poorly utilized, with the consequence of needing to add exogenous phosphorus to meet the birds’ physiological requirements. Probiotic fermentation is an economically feasible processing technology to lessen anti-nutritional components and augment the nutritional quality of plant-based protein meals, including SBM [3]. Adding probiotics during fermentation enhances the concentration of probiotics, enzymes and metabolites and may change some compounds into more effective components [4]. Nevertheless, in order to improve the quality of fermented products, the upgrading of such technology does not stop. The double-stage fermentation process, using two fermenters instead of one, provides an attractive avenue to enhance the degradation of anti-nutritional factors, with a high product yield and quality. In the fungal stage of fermentation, *Aspergillus* is the most commonly used species, due to its ability to secret enzymes such as hemicellulases, hydrolases, pectinases, phytase, amylase, protease, lipases, and tannases [5]. Regarding bacterial fermentation, several *Lactobacillus* species and *Bacillus subtilis* are ideal choices [6]. Double-stage fermentation of SBM using probiotic bacteria and fungi had a great benefit not only to augment native SBM composition and its nutritional value [7], but also to enhance the nutrient utilization and bioavailability with fungal enzymes and bacterial metabolites. Furthermore, bacterial and fungal fermentation can result in efficient degradation and elimination of anti-nutritional factors such as phytates, mycotoxins, trypsin inhibitors, oligosaccharides and, in SBM, also improving the small-sized peptides and essential and non-essential amino acids [8]. The nature of processed poultry feed ingredients can modify the expression of most important genes related to nutrient transportation [2,9]. From this point of view, promoting nutrient transport via transporter-encoding gene upregulation can result in faster nutrient entry into the intestinal epithelium and, afterward, into the blood circulation [10]. Moreover, as the absorption of dipeptides is more effective than that of individual amino acids, the transporter activity of PepT1 in the small intestine is enhanced [8]. These dipeptides may be absorbed more promptly than protein-bound amino acids, since they do not have to go through any further digestion [11]. It is identified that transporters of cationic amino acid (CAT1 and CAT2) and peptide (PepT1 and PepT2) in the intestinal epithelium are associated with the capacity of nutrient absorption [2,12,13,14]. Moreover, estimating phytase activity in fermented SBM, which reflects the efficiency of phosphorus ileal digestibility in broiler chickens, necessitates more exploring. To our knowledge, there are scarce or no data considering the effect of two-stage fermentation of SBM on the expression of genes controlling main protein transporters. In addition to investigating this, the existing study explored the impact of feeding two-stage fermented SBM on the performance and the gene expression of protein nutrient transporters of broiler chickens and investigated to which extent the fermentation products were able to release and enhance available phosphorus.

## 2. Materials and Methods

The bird rearing practices and sampling collection procedures were performed according to the instructions and rules of the Institutional Animal Care and Use Committee of the Faculty of Veterinary Medicine at Zagazig University (ZU-IACUC/2/F/322/2022).

### 2.1. Fermentation and Characterization of Soybean Meal

#### 2.1.1. Fermentation Process of Soybean Meal Was Carried out in Two Stages

The first stage used *Asperigillus oryzae* strain SS_RS-SH (MN894021.1) according to the method of Teng et al. [15] for; activation of *Asperigillus oryzae* the spores were cultivated on potato dextrose agar then incubated at 28 °C for 72 h. Spores were collected in sterile distilled water containing 0.12% Tween 80 and the spore inoculum adjusted to 10^8^ /mL, and it was ready for use. The soybean meal (SBM) used was obtained from local suppliers. At the start, the SBM was steamed at 100 °C for 20 min for sterilization and for increasing moisture content to about 50%, then left to cool to room temperature; after cooling about 10^8^, spores of activated *Asperigillus oryzae* were mixed into each 100 g of steamed SBM and incubated at 37 °C for 24 h.

The second stage of fermentation we used *Bacillus subtilis* SB102 (CGMCC1.769) using the directions of Zhang et al. [16] for preparation of *B. subtilis*; inoculum was developed in Luria-Bertani (LB) at 37 °C for 48 h. Then, the inoculum was adjusted to 10^8^ CFU /mL using uncontaminated physiological saline. After 24 h fermentation of soybean meal by *Asperigillus oryzae,* an inoculum of *B. subtilis* was mixed with the fermented mixture with 10^8^ CFU/ 100 g and incubated at 37 °C for 36 h.

Finally double-stage fermented soybean meal (DFSBM) was dried at 50–60 °C for 72 h until moisture content reached 10.9%, then stored at −20 °C until use in broiler ration formulation.

#### 2.1.2. Evaluation of DFSBM

Three samples of SBM and DFSBM were evaluated in the following:

##### Chemical Analysis

Samples of SBM and DFSBM was taken for triplicate analysis of DM, CP, EE and Ash according to official method of analysis [17] Table 1.

##### Amino Acids Quantification

The quantitative detection of amino acids was performed by gas liquid chromatography (GLC) as suggested by Mabbott [18]. Briefly, it was through hydrolysis of identified weight of SBM and DFSBM protein using 6 mL HCL in a hot air oven for 20 h at 110 °C. After that, 10 mL of distilled water and a few grams of cation exchange resin were used for solubilizing the hydrolyzed mixture at pH 1.8, then gently shaking for about 10 min. Furthermore, 5 mL of ammonia solution (10%) was added to the prepared mixture, then filtered and evaporated by heating at 40 °C. In the next step, 1 mL methanolic HCL reagent was added to the mixture and dried for 30 min in a hot air oven at 70 °C. Finally, before injection into GLC, 250 μL of methanol was added. GLC conditions were the following: Sample: 1 μL injection; Carrier gas: H2 (flow 6.0 mL/min.); GC: HP 6890, intel temperature, 250 °C; and Column: HP-1 methyl siloxane 30 mm × 1.5 μm. The oven also adjusted for 2 min as initial time with final temperature of 290 °C, at a rate of 8 °C/min.

##### Reduced Sugar Analysis

Reduced sugar was analyzed according to the method of [19]. Briefly, glucose standard solution with concentration of 10–100 µg/mL, sample (1 mL) and acetate buffer (1 mL of 0.05 M at pH 4.8) were mixed thoroughly; after that, 3 mL of Dinitrosalicylic acid reagent was added and then subjected for 5 min to a boiling water bath. After cooling of the mixture at room temperature, the absorbance was quantified at 540 nm with UV–visible spectrophotometer.

##### Measurement of pH

Normal soybean meal and FSBM pH were measured following the direction of [20] as follows: in a blender, 10 g of the sample was mixed in 10 mL of neutralized distilled water. After that, the resultant homogenate was kept for 10 min at room temperature with constant shaking. The value of pH was assessed via an electrical pH meter (PSH-3C, INESA Instrument, Shanghai, China).

##### Phosphorus Analysis

The method for phosphorus analysis was performed according to [21] by using a spectrophotometer. Briefly, samples were dried then ashed, then solubilized in acids (nitric and perchloric) in recommended concentrations. The filtrate of the samples was used for measuring the color absorption at 400 nm.

##### Assessment of Phytase Enzyme Activity

The phytase activity was evaluated by determining the released inorganic phosphorus from sodium phytate following the instruction of animal feed stuff–phytase analysis ISO/DIS 30024 [22]. Briefly, soybean samples were extracted by using 0.2 M citrate buffer, at pH 5.3 at room temperature for 30 min. Then samples were filtrated, and the supernatant oily layer was removed. Finally, clear filtrate was used for phytase activity evaluation. The single unit of phytase activity was well-defined as the quantity of phytase that is needed for liberating 1 μg of inorganic phosphorus at 37 °C in 1 min and pH 5.3, measured as (FTU/Kg).

##### Determination of Trypsin Inhibitor

Trypsin inhibitors were extracted in 0.01 N NaOH according to Kakade, Rackis, McGhee, and Puski [23] (1974), and the activity was assayed according to the method of Hamerstrand, Black, and Glover [24].

### 2.2. Birds, Diets and Experimental Design

A total of 200 Ross one-day old broiler chicks (ROSS 308), with initial body weight of 44.2 ± 1, were procured from a commercial hatchery. Chicks were weighed and randomly divided into four treatment groups with 5 replicates pens of 10 birds. The birds were raised in battery cages in an environmentally controlled system. During the 1st week of age, the room temperature was primarily adjusted to 33 °C and then gradually declined until reaching 21 ± 1 °C, and humidity was kept around 60% during the whole experimental period; also the stocking density was 10 birds/M^2^. The experimental diets were formulated as three stages: starter (d 1–10), grower (d 11–24) and finisher (d 25–40), with nutrients that cover the requirements of Ross broiler nutritional specification of ROSS [25]. Clean water and feed were allowed all the time. The birds were offered a basal diet with [0(control), 25%, 50%, and 100%] replacement of normal soybean meal by DFSBM. The percent of feed ingredient and chemical composition of the experimental diets are listed in Table 2. The proximate chemical analysis of the feed ingredients was made according to [17]. Experimental diet amino acids analysis was performed by gas liquid chromatography (GLC) as suggested by Mabbott [18]; additionally, dietary phytase activity was measured according to [22] using the ISO/DIS 30024, as mentioned in soybean meal characterization, as listed in Table 3.

### 2.3. Growth Performnce Attributes and Digestibility Trial

The overall body weight and total feed intake (FI) were weighed weekly, to evaluate FI, body weight gain, and feed conversion ratio (FCR) for the entire experimental period (d 1–40).

### 2.4. Sample Collection and Analytical Procedures

Five birds from each group at the end of the experimental period were randomly selected, weighed, and slaughtered.

For serum biochemical measurement, 3 mL blood sample was collected from each bird (n = 5), then centrifuged for 15 min at 2000 rpm, then kept at −20 °C until further biochemical assessment.

For plasma immunological assessment, 3 mL blood sample was collected in EDTA-containing tubes (n = 5), then centrifuged for 15 min at 3000 rpm. Clear plasma samples were kept at −20 °C until further analysis.

For digestive enzyme analysis, pancreatic samples were direct collected from slaughtered birds (n = 5), homogenized in ice cold phosphate-buffer saline, then centrifuged for 15 min at 3000 rpm. The supernatant was collected into Eppendorf tubes and kept at −70 °C until analyzed.

For antioxidant analysis, liver tissues were homogenized in ice cold phosphate- buffer saline (20 mL; pH 7.4) and glycerol (20 mL; 20%), then centrifuged for 15 min at 3000 rpm. The supernatant was collected into Eppendorf tubes and kept at −70 °C until analyzed.

For meat chemical and amino acids composition, meat samples (250 g) were collected from breast muscles then freeze-dried, then stored at −20 °C until chemical analysis.

For molecular analysis, jejunal samples (I cm of distal jejunum just before the Meckel’s diverticulum separated) were collected, and digesta was evacuated from it, then washed with PBS (NaH_2_PO_4_, 1.47 mmol/L; Na_2_HPO_4_, 8.09 mmol/L; and NaCl, 145 mmol/L) 3 times, and preserved in Trizol reagent at −80 °C until analysis of amino acids- and peptide transporter-encoding genes.

### 2.5. Nutrient Digestibility and Pancreatic Digestive Enzymes

The apparent nutrient digestibility was established with chromic oxide. At the end of the experimental period, chromic oxide was mixed into finisher diets at the rate of 5 g/kg diet. After that, excreta from each replicate cage were assembled every 8 h for seven subsequent days for analyzing crude protein, dry matter, ether extract and crude fiber according to [17]. The content of chromic oxide in excreta and diets was evaluated spectrophotometrically following acid digestion, corresponding to Short et al. [26]. For measuring phosphorus and calcium apparent ileal digestibility, ileum content samples were collected from slaughtered birds and evacuated into clean falcon tubes. Feed and ileal digesta samples were dried, ashed and solubilized in acid solutions, then filtrated and used for calcium evaluation by atomic absorption spectrophotometry according to [27], while phosphorus was measured color metrically according to [21].

The nutrient apparent digestibility coefficient was assessed in accordance with McDonald, P. [28].

Apparent nutrient digestibility = 100 − [100 × (the content of indicator in diet/indicator content in feces × nutrient content in feces/Nutrient content in diet].

For assessing pancreatic digestive enzymes, pancreatic supernatant-extracted samples were used for assessment of digestive enzyme activities. The amylase activity was measured using the method of [29], while the activity of lipase was determined using the method described by [30]. The activity of pancreatic trypsin was activated by a pre-incubation period with 0.08 units of Enterokinase (Sigma Aldrich, nc.St. Louis, MO, USA) for 30 min. After the activation, trypsin was determined at 37 °C according to the method of [23].

### 2.6. Serum Biochemical Analysis

Serum aspartate aminotransferase (AST) and alanine aminotransferase (ALT)), creatinine, urea, uric acid, total cholesterol (TC), triglycerides (TGs), high-density lipoprotein cholesterol (HDL-C), low-density lipoprotein cholesterol (LDL-C), and very-low-density lipoprotein cholesterol (VLDL-C) were determined using commercial diagnostic kits (Spinreact Co., Santa Coloma, Spain).

### 2.7. Assesment of Liver Antioxidant

The extracted supernatant of liver samples was used for measuring the following antioxidant markers [31]. Malonaldehyde (MDA) level was assessed by using standard kits (NWLSS™ MDA assay kits NWK-MDA01). The test principle was based on the reaction of MDA with thiobarbituric acid (TBA), forming an MDA-TBA2 then measuring spectrophotometric light absorption at 532 nm, following the method of Placer et. al. [32]. Superoxide dismutase (SOD) was analyzed using kits (NWLSS™ Superoxide dismutase activity assay NWK-SOD02, Northwest Life Science Specialties, LLC, Washington, USA) following instructions described by [33]. Catalase (CAT) was measured spectrophotometrically according to Cowell, D., A. Dowman, R. Lewis, R. Pirzad and S. Watkins [34] using the kits (NWLSS™ Catalase activity assay kit protocol NWK-CAT01). Glutathione peroxidase (GSH-Px) was measured using commercial assay kits (Glutathione Peroxidase Assay Kit, Colorimetric, ab102530; Cambridge Biomedical Campus, Cambridge, CB2 0AX, UK) consistent with the method of [35].

### 2.8. Chemical and Amino Acids Copmostion of Breast Meat

The dry matter, crude protein, fat and ash content of breast and thigh meat were analyzed according to AOAC [17]. Ca and P retention levels were determined using atomic absorption spectrophotometer Sunostk, SBA733 Plus. Colorimetric methods were used for calcium [36] and phosphorus [37] using ICPAES (Thermo Sci, Model: iCAP6000 series) following the manufacturer’s instructions; absorbance was measured at 585 nm and 460 nm for Ca and P. Breast muscle amino acids analysis was performed by gas liquid chromatography (GLC) as suggested by Mabbott [4], as mentioned in soybean meal characterization.

### 2.9. Cecal Bacterial Count and Duodenal pH

Cecal content samples were collected directly after slaughter, weighed and placed with sterile phosphate buffered saline (PBS, pH 7, 100 mM NaCl) in proportion 1: 10 (w/v) and pummeled for 2 min in a Stomacher. Ten-fold serial dilutions in sterile buffer peptone broth were performed as per the method described by [38], up to 10^7^, and aliquots (0.1 mL) of each dilution were pour-plated in MRS agar (Oxoid), MacConkey agar (Oxoid), and Tryptone Soy Agar (TSA, Oxoid) to determine the main representative cultivable lactic acid microbiota (LAB), coliforms, and total bacterial counts, respectively. The plates were incubated at 37 ± 1 °C for 24 h for coliform counts, and at 30 ± 1 °C for 24 h and 48 h for total bacterial and LAB counts, respectively. The results were expressed as the Log number of colony-forming units per gram (wet weight) of cecal content (Log CFU/g). The duodenal pH was measured by directly inserting the pH meter probe into duodenal content following the steps of [39].

### 2.10. Nutrient Transporter-Encoding Genes Real-Time PCR

Total mRNA from jejunum samples was extracted by Trizol reagent (TaKaRa Biotechnology Co. Ltd., Dalian, Liaoning, China). The extracted RNA was treated with RNeasy Mini Kit (Qiagen, Cat. No. 74104) corresponding to the guidelines of the manufacturer. The amount and pureness of the total RNA were detected by NanoDrop ND-8000 spectrophotometer (Thermo Fisher Scientific, Waltham, MA, USA). Corresponding DNA (cDNA) gained by reversal-transcription of the isolated RNA samples utilizing RevertAidTM H Minus kits (Fermentas Life Science, Pittsburgh, PA, USA). One μL of this cDNA was mixed with 2x maxima SYBR Green PCR mix (12.5 μL) and RNase free water (10.5 μL); later 0.5 μL of each forward and reverse primer for the selected genes were added. The sequences of the primers of genes encoding amino acids and peptides transporters are depicted in Table 4. Glyceraldehyde-3-phosphate dehydrogenase (GAPDH) was employed as control gene.

### 2.11. Statistical Analysis

The analysis of data was computed by means of the general linear model (GLM) process of SPSS (one-way ANOVA), after validating the homogeneity among experimental groups using Levene’s test and normality utilizing Shapiro–Wilk’s test. Tukey’s was utilized to check for significant variations among the mean values. The entire results were expressed as the standard error of the mean (SEM), and the statistical significance was set at *p* < 0.05. Data concerning cecal CFU were transferred to log^10^ CFU numbers prior to analysis. For molecular gene expression, the fold change was estimated as follows: (B-A)/A where the lowest possible value is A and the ultimate value is B. Relative fold changes in the expression of reference genes were estimated by the 2^−ΔΔCt^ method.

## 3. Results

### 3.1. Growth Performance, Nutrient Digestibility and Digestive Enzyme Acivity

The overall growth performance traits, nutrient digestibility and pancreatic digestive enzyme activities are shown in Table 5. The final body weight and body weight gain were significantly improved (*p* > 0.05) in groups fed 50 and 100% DFSBM when compared with the control group. Moreover, the lowest (*p* > 0.05) feed intake was observed in the group fed 25% DFSBM. The feed conversion ratio was improved (*p* > 0.05) with gradual increase in DFSBM levels in broiler diets. Regarding the nutrient digestibility, the groups fed 50 and 100% DFSBM as a replacement for SBM showed the significant highest (*p* > 0.05) dry matter, crude protein and fat digestibility. Moreover, replacing SBM with DFSBM had an improved impact on ileal availability of phosphorus and calcium, simultaneously with increasing DFSBM level in diets. Concerning the activities of pancreatic digestive enzymes, the increase in DFSBM levels in broiler diets was accompanied by a significant (*p* > 0.05) increase in the activities of pancreatic amylase, lipase and trypsin.

### 3.2. Serum Biochemical Parameters

Data regarding the influence of dietary replacement of SBM with DFSBM on serum biochemical parameters are shown in Table 6. The levels of serum liver function tests (AST and ALT) were not affected by the replacement of SBM with DFSBM. Regarding kidney function tests, creatinine levels in broiler serum recorded a significantly (*p* > 0.05) higher level in groups fed 50 and 100% DFSBM than the control and 25% DFSBM-fed groups. Meanwhile, urea and uric acid levels expressed a non-significant (*p* > 0.05) variation among experimental groups. The modulation in serum lipid profile was simultaneous with increased levels of DFSBM in broiler feed, as the total cholesterol, TAGs, VLDL-C and LDL-C levels were significantly lowered (*p* > 0.05) with the increased level of DFSBM in broiler diets; on the contrary, the HDL-C levels were increased.

### 3.3. Hepatic Antioxidant Potential

The effects of dietary replacement of normal SBM with DFSBM on hepatic antioxidant capacity are shown in Table 7. Concerning the activities of hepatic SOD, CAT and GSH-PX acivities, inclusion of higher levels of DFSBM was related with an improvement of their activities. In contrast, the group fed DFSBM at the level of 100% showed the significantly lowest (*p* > 0.05) MDA levels.

### 3.4. Chemical and Amino Acids Copmostion of Breast Meat

The chemical and amino acid composition of broiler breast muscles is shown in Table 8. Dry matter of breast meat was not affected (*p* > 0.05) by dietary substitution of DFSBM. Moreover, crude protein, Ash, calcium and phosphorus content of breast muscle were significantly elevated (*p* > 0.05) with the elevation of DFSBM levels in the diets; however, the ether extract content of the breast meat was significantly lowered (*p* > 0.05) with higher levels of DFSBM in the feed. Regarding the amino acids composition of breast muscles, most of their contents were not significantly affected (*p* > 0.05) by dietary replacement, except for phenyl alanine, which showed a significant lower (*p* > 0.05) content in groups fed 50 and 100% DFSBM, compared to the control group. Additionally, the aspartic and glutamic acid and tyrosine contents of breast muscle were significantly increased (*p* > 0.05) in groups fed with higher levels of DFSBM, with no significant differences detected among the other non-essential amino acids.

### 3.5. Cecal Bacterial Count and Duodenal pH

The cecal coliform, lactobacillus and total bacterial counts and duodenal pH are presented in Table 9. The cecal total bacterial count was significantly higher in DFSBM groups than control. The population of coliforms in cecal samples were significantly decreased (*p* < 0.05) with increasing levels of DFSBM in a level-dependent manner. The cecal fermentative lactobacilli count was significantly increased with increasing the inclusion level of DFSBM when compared to the control group (*p* < 0.05). The duodenal pH was significantly lowered (*p* < 0.05) with increased DFSBM levels when compared to control.

### 3.6. Gene Expression of Protein-Related Nutrient Transporters

The changes in mRNA expression of LAT1, PEPT1, PEPT2, CAT1 and CAT2 genes in response to substitution of DFSBM are illustrated in Figure 1. The group fed DFSBM at the level of 100% exhibited the most prominent upregulation of LAT1 gene. Regarding PEPT1 and PEPT2 gene expression, the fermentation of SBM upregulated (*p* > 0.05) their expression levels, which reached their peak in DFSBM at the level of 100%. All groups fed DFSBM showed a significantly higher (*p* > 0.05) expression of CAT1, compared to the control group. The higher expression (*p* > 0.05) of CAT2 gene was detected in the groups fed DFSBM at the level of 50 and 100% when compared with control group.

## 4. Discussion

Fermented soybean meal (FSBM) attains extensive concern as a functional feed owing to its promising impact on animals, including poultry. Little is known about the synergistic influence between *B. subtilis* and *A. oryzae* for enhancing SBM quality utilizing a double-stage fermentation technique. In the current study, DFSBM not only exerted beneficial impacts on the quality of fermented product, but also supplied the gastrointestinal tract with many benefits, which in turn boosted gut health and growth performance of broiler chickens. The data considering the effects of DFSBM on broiler performance still need more investigation to support its use as a total replacement for SBM, as the quality of fermented feed depends to a large extent on the microbiological load and enzymatic activity occurring during fermentation [40]. Therefore, subjecting SBM to double-stage fermentation using the *A. oryzae* strain then *B. subtilis* has been found to improve its chemical and probiotic load. These data are in agreement with Shi, C., Y. Zhang, Z. Lu and Y. Wang [41], who claimed that microbial fermentation is considered to be an effective tool to provide probiotics and improved nutritional quality through increasing CP and reducing CF and fat contents in fermented feed. Furthermore, higher concentration of protein contents may to some extent be attributed to the reduction in carbohydrate and fat contents over fermentation [42]. Fermentation significantly elevated small-sized peptides (<15 kD) in FSBM [43] and degraded long-chained proteins. Moreover, using *B. subtilis* as probiotic fermented additives enhanced growth and proliferation of lactic acid bacteria (LAB) and in turn reduced pH and therefore prevented the growth of pathogenic bacteria [3]. Notably, the growth rate and FCR were improved, especially in the group fed 100% DFSBM as a total replacement for SBM. The higher concentration of probiotic bacteria as total *lactobacilli* and *B. subtilis* after fermentation can reflect this better performance of broiler chickens. The higher growth performance of groups fed DFSBM could be also attributed to decreasing the level of trypsin inhibitors, which is the main anti-nutritional factor in raw SBM (decreased from 19.96 in raw SBM to 5.43 in DFSBM). Another advantage that was detected after fermentation with *Aspergilli orzae* was the elimination of a higher percentage of phytate, offering a high-quality protein source for feed with highly available phosphorus, which was in agreement with Ilyas, A., M. Hirabayasi, T. Matsui, H. Yano, F. Yano, T. Kikishima, M. Takebe and K. Hayakawa [44]. Similarly, alteration in nutritional and anti-nutritional characteristics of soybean meal by solid-state fermentation with *B. subtilis* and *A. oryzae* was detected and proved the increased CP content and quality of amino acids [45]. Former studies revealed that increased crude protein content after fermentation was due to an increased microbial biomass composed primarily of protein [46]. Additionally, the increase in crude protein percent in DFSBM was mainly attributed to the microbial fermentation of carbohydrates and fat and using them as energy sources [47]. Microbial fermentation can result in hydrolysis of protein and enhanced liberation of free AAs; therefore, the resulting DFSBM has a significantly higher total free AA content compared to SBM [47]. Herein, the crude protein content of DFSBM had increased by 3.67%, with an increase of 12.81% of total essential amino acids and 10.10% in total nonessential amino acids compared to non-fermented soybean meal. Moreover, total free amino acids were increased 5.88-fold in double-stage fermented SBM. The fermentation of soybean meal by *Bacillus spp.* had improved free amino acids, which may be attributed to the higher exogenous protease enzyme activity, secreted through microbial fermentation [48]. In concurrence with the current study, Zhang et al. [16] reported improvement of total crude protein, total essential and nonessential amino acids and total free amino acids. Similar data obtained by Gerham et al. [49] for fermented soybean meal showed it higher total essential, nonessential and free amino acids that lead to improve these amino acids ileal and standard digestibility in growing swine. Phytate is difficult to digest for poultry due to lack of phytase synthesis capacity in their gastrointestinal tracts. Phytate can also bind to many minerals such as phosphorus, other nutrients and digestive enzymes, leading to decreased nutrient digestibility and enlarged nutrient excretion in faces [50]. Although phytate can be partially digested by some phytase-producing gut bacteria, it had limited activity due to small amounts of secreted phytase [51]. Therefore, solid-state fermentation of soybean meal by *A. oryzae* that produce high amounts of exogenous phytase may improve phosphorus availability and other nutrient digestibility [52]. Moreover, adding probiotic feed additive that have the ability to secret phytase enzyme can improve phytate utilization with better phosphorus bioavailability. In this study, phytase activity in DFSBM had been increased by the double-fermentation process and could be attributed to the presence of *A. oryzae* that has the ability to degrade phytate during fermentation via secreting phytase enzyme [53]. Phosphorus is a crucial mineral, since it plays a main role in numerous body functions and in accretion in the skeleton, together with other minerals such as Ca. High phosphorus in poultry feces can also cause environmental complications [45]. In our study, the phosphorus and calcium levels were increased in soybean meal with applying solid-stage fermentation using *A. oryzae* simultaneously with higher phosphorus and calcium ileal digestibility and higher muscle retention in groups fed DFSBM, dependent on replacement level. This interpretation is in concurrence with the study of Chen et al. [52], who reported that double-stage fermentation of SBM achieved better phytic acid degradation. Moreover, improved broiler performance, nutrient digestibility and phosphorus retention were attributed to increased phytase enzyme supplementation in their diets [54]. In addition, the reduced crude fiber in DFSBM can be attributed to production of carbohydrases (cellulase and xylanase) that degrade fiber from *A. oryzae,* which is in agreement with [53]. Additionally, in the current study, the digestibility of DM, CP, calcium and fat was improved, especially at complete substitution of SBM by DFSBM. Similarly, fermentation has also been proven to enhance nutrient digestibility of organic matter, crude protein, fat and calcium [55]. Moreover, adding different fungi during fermentation can be combined simultaneously or sequentially to reduce anti-nutritional factors such as phytate and therefore improving overall digestibility of amino acids, fiber and minerals of the fermented feed. The current study demonstrated that the activities of digestives enzymes (lipase, amylase and trypsin) responsible for digestion and nutrient transport were increased with increasing levels of DFSBM. These advantages could be related to probiotic properties of DFSBM that led to improving the intestinal absorptive surface area and secretion of brush border enzymes, thus achieving boosted digestion and absorption, which is in line with Chiang, G., W. Lu, X. Piao, J. Hu, L. Gong and P. Thacker [56]. Likewise, fermented feeds elevated pancreatic *AMY2A* and *CCK* expression, which in turn increased the secretion of pancreatic amylase and cholecystokinin in broilers [40]. Additionally, increased trypsin and total protease activities in the duodenum and jejunum were noticed after feeding pigs with FSBM [57]. This improvement in protein digestibility could be due to the degradation and reduction in large-sized proteins in FSBM. Peptide and amino acids’ transporters in the small intestines of chicks are controlled by a range of factors, such as development of intestinal dietary protein quantity and quality and genetic selection [58,59]. Dietary protein quality differences can regulate the mRNA expression of nutrient transporters in the bird’s small intestine [58], indicating that the mRNA expression levels of PepT1 and CAT1 decreased with reduced dietary nutrient density [60] In this study, the genes regulating the expression of amino acids’ transporters were upregulated more prominently in the group fed 100% DFSBM. Our observation is in harmony with that of Liu [61], who observed that broilers fed the high-protein diet expressed higher mRNA levels of PepT1 in jejunum than those fed a low-protein diet. Moreover, the higher expression of amino acids’ transporters after feeding on DFSBM at 100% replacement for SBM could result from higher concentration of small peptides, and amino acids could result from fermentation, which upregulated the expression of amino acids’ carriers. In accordance, small peptide expression levels and amino acids’ carriers can imitate the absorption status of small peptides and amino acids in the small intestines of birds, which are directly affected by the substrate concentration and elevated with the elevation of substrate concentration [62]. Following double-stage fermentation of soybeans, their protein was hydrolyzed into small molecules of peptide and amino acids by fermentation enzymes with a consequence of higher concentration of dipeptide, tripeptide and neutral and free amino acids in the intestine, which is a main factor promoting the transcription of CAT1, CAT2, PepT1 and PEPT-2 genes and increasing their expression. In this study, higher breast muscle content of crude protein and some amino acids such as lysine, phenylalanine, aspartic acid, glutamic acid and tyrosine was noticed with increased replacement level with DFSBM. This improvement may be attributed to the higher crude protein of DFSBM, higher free amino acids, improved crude protein and amino acid digestibility, simultaneously with higher expression of protein transporter genes, which all caused higher amino acid retention in breast muscles. In agreement with our results, Guo et al. [63] reported that inclusion of FSBM in broiler chicken diets improved breast muscle meat quality and elevated the retention of some free amino acids such as glutamic acid, leucine and isoleucine. In agreement, Gerham et al. [49] described an improved amino acid digestibility in swine by the addition of FSBM to their diets. The increase in breast muscle amino acids may have resulted from its response to higher availability, digestion and absorption of these amino acids in feed, which improved due to the fermentation processes [64]. Regarding the serum metabolites, replacing soybean meal with DFSBM did not affect liver and kidney function markers, while serum lipid profile was improved through decreasing cholesterol and TAG and elevation of HDL-C concentration. The decrease in cholesterol level in serum may be due to the lactic acid bacteria that are known to lower serum cholesterol and TAG [65] by improving the excretion of bile acids in the feces [66] and hindering 3-hydroxy-3-methyl-glutaryl-CoAreductase, which is a key enzyme in cholesterol formation [67]. The increase in HDL-C in the serum of broilers fed FSBM may be due to an elevation of the antioxidant markers, thus decreasing the lipid oxidation [65]. In the current study, the measured antioxidant markers such as hepatic SOD, CAT and GSH-PX were improved, while MDA levels such as lipid peroxidation markers were decreased with the increasing level of DFSBM in their rations. In line with our results, Luo et al. [68] showed that FSBM had increased muscle CAT and T-AOC in finishing pigs. Furthermore, the addition of FSBM by *Bacillus spp.* to broiler rations had decreased lipid oxidation and stimulated antioxidant functions [69]. The gut microbiota is very crucial to provide normal gastrointestinal and normal digestion of nutrients [70]. Fermentation of SBM using *Bacillus subtilis* increased LAB and *Bacillus* counts in soybean meal. Replacing SBM with DFSBM performed as a probiotic source in broiler feed [63]. Numerous studies described that addition of different types of probiotic bacteria to broiler feed decreases the pathogenic strains and increases *Firmicutes* in intestinal lumen [71,72]. In the existing research, the inclusion of DFSBM in broiler rations had improved gut microbiota and tended the microbial population to high lactic acid bacterial colonization in broiler cecum. Moreover, the direct source of probiotic bacteria from the fermentation process, the acidic pH and organic acids in the DFSBM, increased the intestinal acidity, causing more proliferation of lactic acid bacteria in broiler cecum and improved intestinal integrity [73]. In agreement, numerous studies [1,74] described that fermented feed such as FSBM exhibited probiotic effects by decreasing coliform counts and elevating LAB counts in Japanese quail, while in weaned piglets, fermented soybean meal had increased gut *Firmicutes* and decreased *Bacteroidetes* and *Proteobacteria* [75].

## 5. Conclusions

Double-stage fermentation of soybean meal using fungi *Asperigillus oryzae* and bacteria *Bacillus subtilis* improved its protein, amino acid content and phytase activity, simultaneously with decreasing fiber and trypsin inhibitor content compared to SBM. Replacement of SBM with DFSBM in broiler can be performed up to 100% of SBM, with favorable outcomes for the body weight gain, feed conversion, nutrient and minerals digestibility and absorption, digestive enzymes (lipase, amylase and trypsin), amino acids and peptide transporters (LAT1, CAT-1, CAT-2, PepT-1 and PepT-2). In addition, inclusion of DFSBM improved blood immune response, liver antioxidant status, and breast muscle nutritional value and lowered serum total cholesterol. Moreover, DFSBM replacement decreased duodenal pH to acidic and increased probiotic lactic acid bacteria in broiler intestines.

## Figures and Tables

**Figure 1 animals-13-01030-f001:**
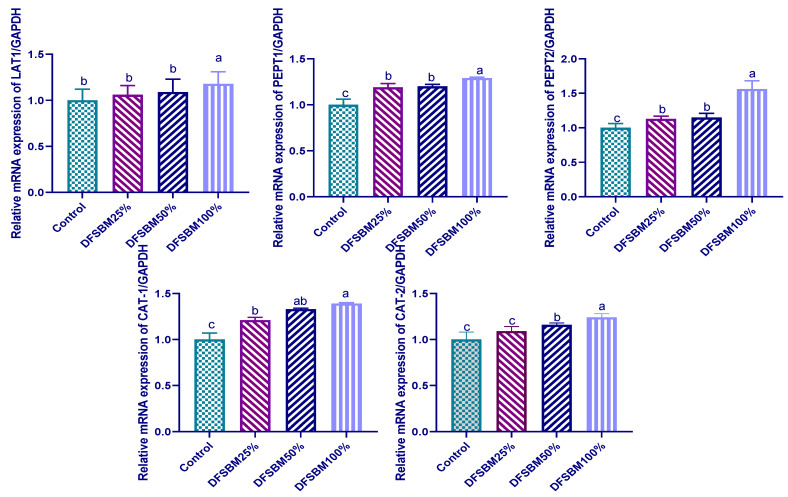
Impact of dietary substitution with different levels of double-fermented soybean meal on amino acids and peptide transporters of broiler chickens. Each value represented mean ± SEM. LAT1: Na+-independent cationic and neutral amino acids; CAT-1: Na+-independent cationic amino acids transporter-1; CAT-2: Na+-independent cationic amino acids transporter-2; PepT-1: Oligopeptide transporter-1; PepT-2: Oligopeptide transporter-2; DFSBM: double-fermented soybean meal. a–c Means with numerous superscripts in the same row diverge significantly (*p* < 0.05).

**Table 1 animals-13-01030-t001:** Chemical analysis of soybean meals used in experimental diets.

**Parameters**	**Normal SBM**	**DFSBM**	**SEM**	***p*-Value**
DM%	87.50 ^b^	89.11 ^a^	0.36	<0.001
CP%	47.5 ^b^	49.23 ^a^	0.39	<0.001
EE%	1.48 ^a^	0.913 ^b^	0.13	<0.001
Ash%	6.07 ^b^	7.12 ^a^	0.24	<0.001
CF%	3.69 ^a^	3.18 ^b^	0.16	<0.001
Reduced suger (mg/g)	165 ^a^	30 ^b^	0.58	<0.001
Phosphorus%	0.67 ^b^	0.93 ^a^	0.06	<0.001
pH	7.37 ^a^	5.25 ^b^	0.48	<0.001
Phytase activity (FTU/kg)	26 ^b^	883.33 ^a^	0.88	<0.001
Trypsin inhibitor level mg/g (dry weight basis)	19.96 ^a^	5.29 ^b^	3.28	<0.001
Essential amino acids (% of DM)				
Argenine	3.45 ^b^	3.66 ^a^	0.05	<0.001
Histidine	1.24 ^a^	1.15 ^b^	0.02	<0.001
Isoleucine	2.25 ^b^	2.44 ^a^	0.04	<0.001
Leucine	3.78 ^b^	4.01 ^a^	0.05	<0.001
Lysine	3.10 ^b^	4.34 ^a^	0.28	<0.001
Methionine	0.67 ^b^	0.81 ^a^	0.03	<0.001
Phenylalanine	2.47 ^b^	2.80 ^a^	0.07	<0.001
Threonine	1.82 ^b^	1.88 ^a^	0.01	<0.001
Tryptophan	0.71 ^b^	0.89 ^a^	0.04	<0.001
Valine	2.35 ^b^	2.68 ^a^	0.07	<0.001
Total essential amino acids	21.86 ^b^	24.66 ^a^	0.63	<0.001
Non-essential amino acids (% of DM)				
Alanine	2.17 ^b^	2.32 ^a^	0.03	<0.001
Aspartic acid	5.40 ^b^	6.32 ^a^	0.20	<0.001
Cystine	0.68 ^b^	0.77 ^a^	0.02	<0.001
Glutamic acid	8.55 ^b^	9.33 ^a^	0.17	<0.001
Glycine	2.03 ^b^	2.12 ^a^	0.02	<0.001
Proline	2.37 ^b^	2.51 ^a^	0.03	<0.001
Serine	2.11 ^b^	2.23 ^a^	0.03	<0.001
Tyrosine	1.72 ^b^	1.98 ^a^	0.06	<0.001
Total nonessential amino acids	25.03 ^b^	27.57 ^a^	0.56	<0.001
Total free amino acids	0.73 ^b^	4.76 ^a^	0.91	<0.001
Probiotic bacteria counts				
Lactic acid bacteria log cfu/g feed	4.25 ^b^	5.70 ^a^	0.32	<0.001
Bacillus spp log cfu/g feed	3.21 ^b^	6.26 ^a^	0.68	<0.001

SBM: soybean meal; DFSBM: double-fermented soybean meal; DM: dry matter; CP: crude protein; EE: ether extract; CF: crude fibre. ^a,b^ Means with numerous superscripts in the same row diverge significantly (*p* < 0.05).

**Table 2 animals-13-01030-t002:** Ingredients constituting experimental diets and their chemical composition (as dry matter).

Ingredients	Starter Rations				Grower Rations				Finisher Rations			
	Control	DFSBM Replacement %			Control	DFSBM Replacement %			Control	DFSBM Replacement %		
		25%	50%	100%		25%	50%	100%		25%	50%	100%
Yellow corn	55.4	55.4	55.4	55.4	58	58	58	58	62.9	62.9	62.9	62.9
SBM, 47.5%	39.14	29.35	19.57	0	35.6	26.7	17.8	0	30.5	22.88	15.25	0
DFSBM	0	9.79	19.57	39.14	0	8.9	17.8	35.6	0	7.63	15.25	30.5
Soybean oil	1.66	1.66	1.66	1.66	2.5	2.5	2.5	2.5	2.5	2.5	2.5	2.5
Calcium carbonate	0.9	0.9	0.9	0.9	0.9	0.9	0.9	0.9	0.9	0.9	0.9	0.9
Calcium dibasic phosphate	2	2	2	2	2	2	2	2	2	2	2	2
Common salt	0.3	0.3	0.3	0.3	0.3	0.3	0.3	0.3	0.3	0.3	0.3	0.3
Premix *	0.3	0.3	0.3	0.3	0.3	0.3	0.3	0.3	0.3	0.3	0.3	0.3
DL-Methionine, 98%	0.2	0.2	0.2	0.2	0.2	0.2	0.2	0.2	0.2	0.2	0.2	0.2
L-Lysine HCL, 78%	0	0	0	0	0.1	0.1	0.1	0.1	0.3	0.3	0.3	0.3
Anti-mycotoxin	0.1	0.1	0.1	0.1	0.1	0.1	0.1	0.1	0.1	0.1	0.1	0.1
**Calculated Composition**												
ME, Kcal/Kg	2979.07	2979.46	2979.27	2979.85	3056.97	3057.15	3057.32	3057.68	3103.94	3104.34	3104.25	3104.55
CP, %	23.29	23.17	23.10	23.32	21.67	21.73	21.80	21.93	19.86	19.92	19.98	20.09
CF%	2.72	2.62	2.67	2.52	2.65	2.60	2.56	2.47	2.58	2.54	2.50	2.42
EE%	4.53	4.42	4.48	4.31	5.41	5.36	5.31	5.21	5.54	5.50	5.46	5.37
Ca, %	0.97	1.02	0.99	1.07	0.96	0.98	1.01	1.05	0.95	0.97	0.99	1.03
Available phosphorus, %	0.67	0.72	0.69	0.77	0.64	0.67	0.69	0.74	0.61	0.63	0.65	0.69
Phytase activity (FTU/kg)	59	368	575	960	54	351	534	917	48	337	502	884

* Supplied per kg of diet: Vitamin D3, 2200 IU; Vitamin A, 12 000 IU; Vitamin K3, 6.25 mg; Vitamin E, 26 IU; Vitamin B1, 3.90 mg; Vitamin B6, 1.5 g; Vitamin B12, 0.31 mg; Pantothenic acid, 18.8 mg; Niacin, 30 mg; Vitamin B2, 6.6 mg; Biotin, 0.5 mg; Folic acid, 1.25 mg; Se, 0.20 mg; Fe, 60 mg; Cu, 6 mg; I, 1 mg; Mn, 60 mg; Co, 1.5 mg; Zn, 60 mg; Choline chloride, 500 mg. ME, metabolic energy; CP, crude protein; EE, ether extract; CF, crude fiber; Ca, calcium; P, Phosphorus.

**Table 3 animals-13-01030-t003:** The composition of amino acids of the experimental diets (% of dry matter).

Ingredients	Starter Rations				Grower Rations				Finisher Rations			
	Control	DFSBM Replacement %			Control	DFSBM Replacement %			Control	DFSBM Replacement %		
		25%	50%	100%		25%	50%	100%		25%	50%	100%
Essential amino acids (% of DM)												
Argenine	1.60	1.64	1.62	1.68	1.49	1.50	1.52	1.56	1.33	1.35	1.36	1.39
Histidine	0.62	0.60	0.61	0.58	0.58	0.57	0.56	0.55	0.53	0.52	0.52	0.50
Isoleucine	1.02	1.06	1.04	1.09	0.95	0.97	0.98	1.01	0.85	0.86	0.87	0.90
Leucine	2.00	2.05	2.02	2.09	1.89	1.91	1.93	1.97	1.74	1.76	1.78	1.81
Lysine	1.36	1.60	1.48	1.84	1.33	1.44	1.55	1.77	1.34	1.44	1.53	1.72
Methionine	0.54	0.56	0.56	0.60	0.52	0.53	0.55	0.57	0.49	0.51	0.52	0.54
Phenylalanine	1.17	1.24	1.20	1.30	1.09	1.12	1.15	1.21	0.99	1.01	1.04	1.09
Threonine	0.88	0.89	0.88	0.90	0.82	0.83	0.83	0.84	0.74	0.75	0.75	0.76
Tryptophane	0.31	0.35	0.33	0.38	0.29	0.30	0.32	0.35	0.25	0.27	0.28	0.31
Valine	1.19	1.25	1.22	1.31	1.12	1.14	1.17	1.23	1.02	1.04	1.07	1.12
Non-Essential amino acids (% of DM)												
Alanine	1.09	1.12	1.11	1.15	1.03	1.04	1.05	1.08	0.94	0.95	0.96	0.98
Aspartic acid	2.43	2.61	2.52	2.79	2.26	2.34	2.42	2.58	2.01	2.08	2.15	2.29
Cystine	0.38	0.39	0.39	0.41	0.36	0.37	0.37	0.39	0.33	0.34	0.35	0.36
Glutamic acid	4.09	4.24	4.17	4.39	3.82	3.89	3.96	4.10	3.45	3.51	3.57	3.69
Glycine	0.97	0.98	0.98	1.00	0.90	0.91	0.92	0.93	0.81	0.82	0.83	0.84
Proline	1.35	1.38	1.37	1.40	1.29	1.30	1.31	1.33	1.20	1.21	1.22	1.24
Serine	1.03	1.05	1.04	1.08	0.97	0.98	0.99	1.01	0.88	0.89	0.89	0.91
Tyrosine	0.76	0.81	0.79	0.86	0.71	0.73	0.75	0.80	0.63	0.65	0.66	0.70

DFSBM: double-fermented soybean meal.

**Table 4 animals-13-01030-t004:** Primer sequences of focused genes applied for Q-PCR reactions.

Gene	Gene Full Name	Primer Sequence (5′-3′)	Accession No
*LAT1*	Na+-independent cationic and neutral amino acid	F: CTCTCTCTCATCATCTGGGC R: TCATTCCTGGGTCTGTTGCT	XM_415975
*CAT-1*	Na+-independent cationic amino acid transporter-1	F: ATGTAGGTTGGGATGGAGCC R: AACGAGTAAGCCAGGAGGGT	XM_015277949.1
*CAT-2*	Na+-independent cationic amino acid transporter-2	F: CAAGTCTTCTCGGCTCTAT R: GTGCCTGCCTCTTACTCA	XM_015285435.1
*PepT-1*	Oligopeptide transporter-1	F:TTTCCTTTACATCCCTCTCC R:TCACTTCTACTCTCACTC	NM-204365
*PepT-2*	Oligopeptide transporter-2	F:TGACTGGGCATCGGAACAA R:ACCCGTGTCACCATTTTAACCT	NM_001319028.1
*GAPDH*	Glyceraldahyde -3-phosphate dehydrogenase	F-GGTGGTGCTAAGCGTGTTA R-CCCTCCACAATGCCAA	NM205518

**Table 5 animals-13-01030-t005:** Impact of dietary substitution with different levels of double-fermented soybean meal on overall growth performance, nutrient digestibility and digestive enzymes of broiler chickens.

Parameter	Control	Replacement % with DFSBM			SEM	*p*-Value
		25	50	100		
Overall performance						
Initial weight (g/bird)	44.67	44.33	44.00	43.83	0.23	0.635
Final BWT (g/bird)	2269 ^c^	2293 ^c^	2448 ^b^	2559 ^a^	36.10	<0.001
Total BWG (g/bird)	2224 ^c^	2248 ^c^	2404 ^b^	2515 ^a^	36.20	<0.001
Total FI (g/bird)	3838 ^a^	3679 ^b^	3815 ^a^	3904 ^a^	27.50	0.003
Total FCR	1.73 ^a^	1.64 ^b^	1.59 ^c^	1.55 ^d^	0.02	<0.001
Apparent Nutrient digestibility %						
DM	77.38 ^c^	80.39 ^b^	81.74 ^a^	81.71 ^a^	0.54	<0.001
CP	72.31 ^c^	74.05 ^b^	75.13 ^a^	75.71 ^a^	0.41	<0.001
EE	82.33 ^b^	82.39 ^b^	84.30 ^a^	84.61 ^a^	0.32	<0.001
Phosphorus	50.68 ^d^	53.14 ^c^	59.01 ^b^	63.37 ^a^	1.50	<0.001
Calcium	38.16 ^d^	40.56 ^c^	43.21 ^b^	46.36 ^a^	3.21	<0.001
Pancreatic digestive enzyme activity						
Amylase (u/mg prot)	161.97 ^c^	164.63 ^b^	166.49 ^a^	167.93 ^a^	0.70	<0.001
Lipase (u/mg prot)	403.95 ^c^	405.36 ^bc^	406.80 ^ab^	409.08 ^a^	0.62	0.002
Trypsin (u/mg prot)	483.33 ^b^	483.95 ^ab^	484.86 ^a^	485.06 ^a^	0.24	0.005

DFSBM = double-fermented soybean meal; BW = body weight; BWG = body weight gain; FI = feed intake; FCR = feed conversion ratio; DM= dry matter; CP = crude protein; EE = ether extract. ^a–d^ Means with numerous superscripts in the same row diverge significantly (*p* < 0.05).

**Table 6 animals-13-01030-t006:** Impact of dietary substitution with different levels of double-fermented soybean meal on liver and kidney function tests and serum lipid profile of broiler chickens.

Parameter	Control	Replacement % with DFSBM			SEM	*p*-Value
		25	50	100		
Liver function tests						
ALT, U/L	17.89	17.81	17.57	17.82	0.10	0.740
AST, U/L	17.98	17.70	17.53	17.60	0.16	0.811
Kidney function tests						
Creatinine, mg/dL	0.94 ^b^	1.04 ^ab^	1.10 ^a^	1.08 ^a^	0.02	0.018
Uric acid, mg/dL	4.07	4.07	4.14	4.29	0.07	0.666
Serum lipid profile						
Total Cholesterol, mg/dL	132.44 ^a^	129.54 ^b^	124.90 ^c^	119.89 ^d^	1.45	<0.001
TAGs, mg/dL	62.52 ^a^	57.88 ^b^	55.73 ^c^	50.85 ^d^	1.28	<0.001
HDL-C, mg/dL	89.85 ^c^	92.56 ^b^	94.03 ^b^	98.62 ^a^	0.99	<0.001
LDL-C, mg/dL	30.09 ^a^	25.41 ^b^	19.72 ^c^	11.10 ^d^	2.17	<0.001
VLDL-C, mg/dL	12.50 ^a^	11.58 ^b^	11.15 ^c^	10.17 ^d^	0.26	<0.001

^a–d^ Means with numerous superscripts in the same row diverge significantly (*p* < 0.05). DFSBM: double-fermented soybean meal; ALT: Alanine aminotransferase; AST: Aspartate aminotransferase; TGs: Triglycerides; HDL-C: high-density lipoprotein cholesterol; LDL-C: low-density lipoprotein cholesterol; VLDL-C: very-low-density lipoprotein cholesterol.

**Table 7 animals-13-01030-t007:** Impact of dietary substitution with different levels of double-fermented soybean meal on hepatic antioxidant capacity of broiler chickens.

Parameter	Control	Replacement % with DFSBM			SEM	*p*-Value
		25	50	100		
MDA nmol/mg	0.417 ^a^	0.409 ^b^	0.402 ^c^	0.397 ^d^	0.002	<0.001
SOD u/mg prot	973.34 ^d^	976.74 ^c^	983.92 ^b^	994.33 ^a^	2.44	<0.001
CAT u/g prot	34.37 ^d^	38.51 ^c^	41.70 ^b^	46.55 ^a^	1.35	<0.001
GSH-PX u/g prot	37.07 ^d^	41.87 ^c^	46.19 ^b^	52.11 ^a^	1.68	<0.001

^a–d^ Means with numerous superscripts in the same row diverge significantly (*p* < 0.05). DFSBM: double-fermented soybean meal; IgG: immunoglobulin G; IgM: immunoglobulin M; NO: nitric oxide; MDA: malodialdhyde; SOD: superoxide dismutase; CAT: catalase; GSH-PX: glutathione peroxidase.

**Table 8 animals-13-01030-t008:** Impact of dietary substitution with different levels of double-fermented soybean meal on chemical and amino acids composition of broiler chicken breast muscles.

Parameter	Control	Replacement % with DFSBM			SEM	*p*-Value
		25	50	100		
Chemical composition of breast muscle (% of DM)						
DM	25.66	25.66	25.86	25.66	0.14	0.959
CP	85.60 ^b^	86.07 ^ab^	86.62 ^a^	86.31 ^a^	0.13	0.011
EE	6.53 ^a^	6.26 ^b^	6.11 ^c^	6.00 ^c^	0.06	<0.001
Ash	4.26 ^b^	4.33 ^ab^	4.39 ^a^	4.45 ^a^	0.02	0.008
Calcium	0.20 ^b^	0.21 ^b^	0.22 ^a^	0.23 ^a^	0.004	0.001
Phosphorus	0.89 ^c^	0.91 ^b^	0.92 ^a^	0.93 ^a^	0.004	<0.001
Amino acids composition of breast muscle (% of DM)						
Essential amino acids (% of DM)						
Argenine	1.61	1.61	1.61	1.61	0.003	1.00
Histidine	0.77	0.77	0.77	0.76	0.003	0.482
Isoleucin	1.24	1.23	1.23	1.23	0.003	0.813
Leucine	2.25	2.25	2.25	2.25	0.002	0.752
Lysine	2.44	2.44	2.42	2.60	0.03	0.090
Methionine	0.47	0.46	0.47	0.47	0.002	0.532
Phenylalanine	1.21 ^a^	1.21 ^a^	1.19 ^b^	1.18 ^c^	0.005	0.001
Therionine	0.86	0.87	0.87	0.86	0.002	0.201
Tryptophan	2.52	2.54	2.54	2.54	0.005	0.162
Valine	1.27	1.27	1.27	1.27	0.002	0.951
Non-essential amino acids (% of DM)						
Alanine	1.36	1.37	1.36	1.38	0.003	0.403
Aspartic acid	2.21 ^b^	2.23 ^a^	2.23 ^a^	2.23 ^a^	0.004	0.007
Cysteine	0.28	0.27	0.27	0.28	0.002	0.287
Glutamic acid	3.52 ^b^	3.54 ^a^	3.53 ^ab^	3.54 ^a^	0.003	0.021
Glycine	1.11	1.12	1.11	1.12	0.003	0.802
Proline	0.85	0.85	0.85	0.85	0.002	0.672
Serine	0.81	0.81	0.81	0.81	0.002	0.951
Tyrosine	0.95 ^d^	0.95 ^c^	0.95 ^b^	0.95 ^a^	0.002	0.752

^a–d^ Means with numerous superscripts in the same row diverge significantly (*p* < 0.05). DFSBM ( double-fermented soybean meal); DM (dry matter); CP (crude protein); EE (ether extract).

**Table 9 animals-13-01030-t009:** Impact of dietary substitution with different levels of double-fermented soybean meal on cecal bacterial count, duodenal histomorphology and pH of broiler chickens.

Parameter	Control	Replacement % with DFSBM			SEM	*p*-Value
		25	50	100		
Cecal bacterial count (log10 CFU/g cecal content)						
Total bacterial count	4.54 ^b^	5.32 ^a^	5.47 ^a^	5.51 ^a^	0.12	<0.001
Coliform count	2.59 ^a^	2.11 ^b^	1.93 ^b^	1.36 ^c^	0.13	<0.001
Lactobacillus count	2.89 ^d^	3.19 ^c^	3.88 ^b^	4.46 ^a^	0.18	<0.001
Duodenal pH						
pH	5.67 ^a^	5.12 ^b^	4.85 ^c^	4.69 ^d^	0.11	<0.001

^a–d^ Means with numerous superscripts in the same row diverge significantly (*p* < 0.05). DFSBM: double-fermented soybean meal.

## Data Availability

The data source is available from corresponding author upon request.

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
