# Peer review of "Double-Fermented Soybean Meal Totally Replaces Soybean Meal in Broiler Rations with Favorable Impact on Performance, Digestibility, Amino Acids Transporters and Meat Nutritional Value"

_animals, 2023, doi:10.3390/ani13061030_

Round 1

Reviewer 1 Report

The work is interesting and brings new elements regarding the use of double-fermented soybean meal in broiler chicken nutrition. The purpose of the work is clearly formulated. The material used for the research is sufficient, the research methods have been selected appropriately and described in detail. However, I lack information on the conditions of keeping the chickens: stocking density/m2, air temperature and humidity in experimental rooms and mortality.

The results are presented in 9 tables and a figures. The tables are legible, and the differences between the individual groups have been marked correctly.

The discussion of the results compared to the results of other authors is not very detailed. The publications cited by the authors of the article are properly selected. The conclusions are correct and result from the obtained research results.

Author Response

Thank you very much for your effort and valuable comment for improving the quality of our manuscript and we fellow yours advise and suggestions and the manuscript has been revised

The work is interesting and brings new elements regarding the use of double-fermented soybean meal in broiler chicken nutrition. The purpose of the work is clearly formulated. The material used for the research is sufficient, the research methods have been selected appropriately and described in detail. However, I lack information on the conditions of keeping the chickens: stocking density/m2, air temperature and humidity in experimental rooms and mortality.

Response:

  • Thank you for your valuable comment, conditions were added.
  • During the whole experimental period there was no morality detected in all experimental group except one bird in control group.

The results are presented in 9 tables and a figures. The tables are legible, and the differences between the individual groups have been marked correctly. Response:

  • Thank you for your positive comment.

The discussion of the results compared to the results of other authors is not very detailed. The publications cited by the authors of the article are properly selected. The conclusions are correct and result from the obtained research results.

Response:

  • Thank you for your positive comment.

Reviewer 2 Report

Overall a very good study. Extremely important industry oriented work, hence, congratulations and well done to the whole team. Yet, a little improvement in write up may further enhance the scope of work.  

Author Response

Thank you very much for your effort

Overall a very good study. Extremely important industry oriented work, hence, congratulations and well done to the whole team. Yet, a little improvement in write up may further enhance the scope of work.

  • Thank you so much for your kind words. We really appreciate you taking the time to share your experience with us.

Reviewer 3 Report

The scientific work submitted for evaluation has great merit in terms of content and interest. Research methods were described in great detail, with great care.

The discussion of the obtained results is carried out with reference to correctly selected literature data. The presented conclusions are documented in the results of the investigation obtained.

The scientific work submitted for evaluation has great merit in terms of content and interest.  Research methods were described in great detail, with great care. The discussion of the obtained results is carried out with reference to correctly selected literature data. I consider the selection of literature items to be appropriate. The authors are well versed in the scientific output in the field under discussion. The presented conclusions are documented in the results of the investigation obtained.The conclusions drawn are consistent with the evidence and arguments presented and address the main question posed. I think the work is very valuable in terms of application.  The results of the research carried out will certainly find application in breeding practice.

Author Response

The scientific work submitted for evaluation has great merit in terms of content and interest.  Research methods were described in great detail, with great care. The discussion of the obtained results is carried out with reference to correctly selected literature data. I consider the selection of literature items to be appropriate. The authors are well versed in the scientific output in the field under discussion. The presented conclusions are documented in the results of the investigation obtained.The conclusions drawn are consistent with the evidence and arguments presented and address the main question posed. I think the work is very valuable in terms of application.  The results of the research carried out will certainly find application in breeding practice.

 Response:

  • Thank you so much for your kind words. We really appreciate you taking the time to share your experience with us.